# Aerodynamic Simulations for Floating Darrieus-Type Wind Turbines with Three-Stage Rotors

**Mohamed Amine Dabachi [1,2,*]** , **Abdellatif Rahmouni [1]**, **Eugen Rusu [3]** **and Otmane Bouksour [1]**

1.  Laboratory of Mechanics Production and Industrial Engineering (LMPGI), High School of Technology (ESTC), Hassan II University of Casablanca, Route d'El Jadida, Km 7, Oasis, Casablanca 8012, Morocco; abd.rahmouni@gmail.com (A.R.); bouksour2@gmail.com (O.B.)
2.  Doctoral Studies Center of National High School of Electricity and Mechanics (ENSEM), Hassan II University of Casablanca, Route d'El Jadida, Km 7, Oasis, Casablanca 8018, Morocco
3.  Department of Mechanical Engineering, University Dunarea de Jos of Galati, 800008 Galati, Romania; Eugen.Rusu@ugal.ro
*   Correspondence: ma.dabachi@ensem.ac.ma; Tel.: +212-645785786

**Abstract:** Growing energy demand is causing a significant decrease in the world's hydrocarbon stock in addition to the pollution of our ecosystem. Based on this observation, the search for alternative sorts of energy to fossil fuels is being increasingly explored and exploited. Wind energy is experiencing a very important development, and it offers a very profitable opportunity for exploitation since the wind is always available and inexhaustible. Several technical solutions exist to exploit wind energy, such as floating vertical axis wind turbines (F-VAWTs), which provide an attractive and cost-effective solution for exploiting higher resources of offshore wind in deep water areas. Recently, the use of the Darrieus vertical axis wind turbine (VAWT) offshore has attracted increased interest because it offers significant advantages over horizontal axis wind turbines (HAWTs). In this context, this article presents a new concept of floating Darrieus-type straight-bladed turbine with three-stage rotors. A double-multiple stream tube (DMST) model is used for aerodynamic simulations to examine several critical parameters, including, solidity turbine, number of blades, rotor radius, aspect ratio, wind velocity, and rotor height. This study also allows to identify a low solidity turbine ($\sigma = 0.3$), offering the best aerodynamic performance, while a two-bladed design is recommended. Moreover, the results also indicate the interest of a variable radius rotor, as well as the variation of the height as a function of the wind speed on the aerodynamic efficiency.

**Keywords:** vertical axis; darrieus turbines; floating wind; double-multiple stream tube; three-stage rotors

## 1. Introduction

In the last few years, the world has seen a growing interest in environmental conservation and the engagement of socio-economic development plans based on a global vision of sustainable development. In order to reconcile the energy needs of countries with the requirements of the preservation of the environment, the international energy strategy aims to increase the share of renewable energies. According to the International Renewable Energy Agency (IRENA) [1], "the objective set for the year 2050 is to reach 17% of the global installed offshore wind capacity (6044 GW)".

To achieve this capacity, it is necessary to look for technological solutions for more powerful and efficient wind turbines [2–4]. The offshore wind turbine is considered as one of many solutions, defined as a wind turbine exploiting the energy generated by the wind at sea [5].

Consequently, the use of this type of wind turbine has significant advantages over onshore wind turbines. Indeed, at sea, the winds are particularly strong and stable, which means more power generated. Moreover, the installation of offshore wind turbines on large water bodies significantly reduces visual and noise pollution for residents [6]. There are two types of offshore wind turbines: VAWTs and HAWTs. The vast majority of installed offshore wind power solutions are HAWTs, because of their maximum power coefficient value, approximately up to 50% of the Betz limit (59.3%) [7]. In comparison, the VAWT procures 40%. There are also other theoretical formulas that announce a maximum power coefficient value of about 61.7% or 64% for Darrieus VAWTs. This result is due to the fact that the blades cross the airflow twice, first in the upstream phase of the rotation, and second in the downstream phase [5].

VAWTs can be classified into two main types based on the wind velocity, efficiency desired, and utilization. The first type is a Savonius wind turbine (drag-driven family) that consists of two or more simple semicircular blades. The design of the Savonius rotor makes them unsuitable for large-scale offshore application, but it can be attractive for small-scale application as domestic wind turbines due to good starting characteristics, easy installation, and low cost [8,9]. The second type is Darrieus VAWTs (lift-driven family) that demonstrate good efficiency compared to Savonius wind turbines, because of their simple blade design and lower center of gravity, making them more attractive for offshore applications. The first patent for a modern Darrieus VAWT was deposited by the French inventor Georges Jean Marie Darrieus first in France in 1925, then in the United States in 1931. The patent covered two major configurations: curved non-straight blades (non-SB) and straight blades (SB), and this is illustrated in Figure 1. There are several variations of non-SB, such as cantilevered and guy-wired versions [10,11]. These types of non-SB VAWTs minimize the bending moments in the blades and are more prone to the dynamic stall than SB VAWTs as demonstrated by Scheurich [12]. In addition, SB has several variations: H-rotor, helical H-rotor, tilted H-rotor, articulating H-rotor. SB-VAWT rests better than non-SB-VAWT due to self-regulation, simple rotor geometry, absence of guy wire use, cost, etc.

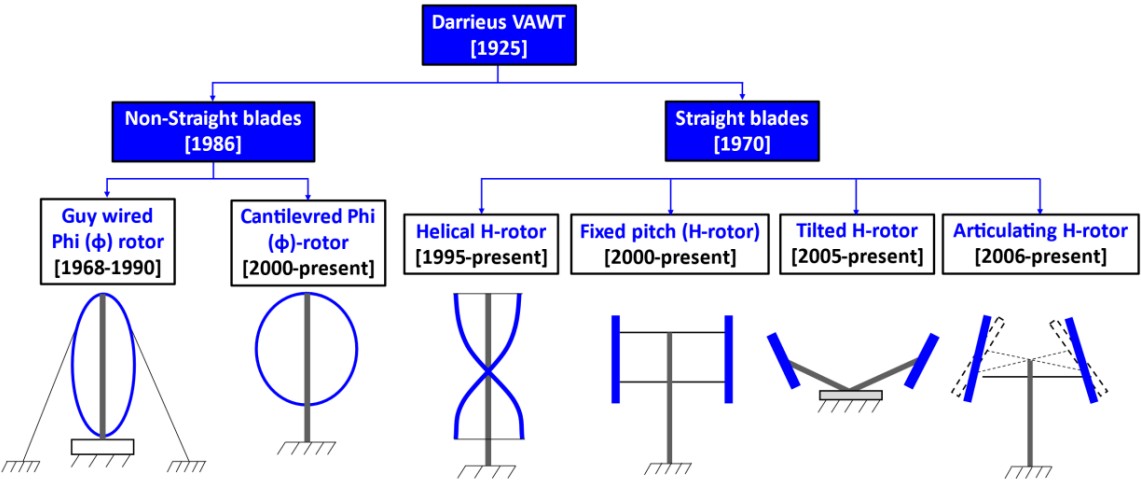

**Figure 1.** Types of Darrieurs VAWT.

Offshore wind turbine goes to deep water. According to a study conducted by the National Renewable Energy Laboratory (NREL) [13], a floating foundation (tensioned-leg platform, spar floater, semi-submersible platform) is more cost-effective than fixed foundations (monopile, tripod, jacket frames . . . ) from a water depth of about 60–100 m. That is why current research on offshore wind turbines is focused on F-VAWTs that have several advantages compared to F-HAWTs, such as: simple mechanism, low center of gravity, no yaw control, withstand high turbulence wind, easier maintenance, cost effectiveness, etc. [14,15]. These benefits have motivated several countries such as the USA, France, Sweden, and Japan to create F-VAWT design projects. Hiromichi et al. [16] developed a new concept

of F-VAWT, called FAWT. This concept used a new mechanism named "bearing swivel rollers" to support the turbine axis and help transform the torque from the turbine to the electric generator. Nakamura et al. [17] has also developed the Savonius keel wind turbine Darrieus concept (SKWID), an improvement of the FAWT concept. In fact, it is a combination between a floating wind turbine and a Savonius water turbine. The aim of the Savonius water turbine is to provide a starting torque to overcome the inertia of the wind turbine, because the VAWT weakness is that it requires some external assistance to initiate the rotation, but Dominy et al. [18] and Hill et al. [19] indicated that self-starting is possible by using symmetrical airfoils under steady wind conditions for a two-blade H-rotor with fixed geometry.

The French company Nenuphar proposed two wind turbines. The first one is VertiWind [20], which has a very high availability rate. The cost of its foundation can be reduced by optimizing the float architecture. The second one is Twinfloat [21], which is built based on two turbines of 2.5 MW, working in a counter-rotating mode, and creating a tunnel effect between them in order to increase electricity production. Other design projects are described in [22–24]. The aim of this paper is to propose a novel design of Darrieus-type straight-bladed F-VAWT with three-stage rotors. A DMST model was adopted in the aerodynamic modeling in order to analyze several critical parameters of VAWT, taking into consideration the turbine solidity, the number of blades, the rotor radius, and the aspect ratio. The remainder of the paper is organized as follows: the proposed design of F-VAWT with three-stage rotors is discussed in Section 2. Section 3 shows general mathematical expressions for the aerodynamic analysis of straight-bladed VAWT. Section 4 is devoted to the DMST model, while Section 5 presents the results and discussion. Lastly, Section 6 concludes the paper.

## 2. Floating Darrieus-Type Wind Turbine with Three-Stage Rotors

In this section, a new design of Darrieus F-VAWT with three-stage rotors is shown in Figure 2. The rotors will have a straight-blade configuration, as it has been shown to demonstrate a higher aerodynamic performance and be self-regulating in all wind velocities in comparison to other possible VAWT configurations [25]. The concept allows having a higher height in order to benefit from the stronger winds. The three rotors of the wind turbine rotate independently around the fixed shaft, unlike conventional VAWTs where the rotor and shaft rotate at the same time, causing the increase of the inertia and the torque applied to the shaft [26], and create problems in the self-starting of the turbine. In addition, by reducing the number of bearings, the axial and radial forces exerted on the shaft are minimized, as well as the related manufacturing cost, compared to the rotating shaft, due to the simple geometric specifications and because our solution does not need much tolerance interval precision [27]. In this design, three permanent-magnet synchronous motors (PMSMs) can be used with different powers depending on the rotor radius, wind velocity, and the height where each rotor is located. The float used is a semi-submersible tri-float type. We are also considering the introduction of connection beams to support the VAWT and increase its stability. Moreover, by using aerodynamic airfoils for turbine supporting arms (or struts), we can avoid the problem of decreasing the aerodynamic efficiency of the turbine. Ahmadi-Baloutaki et al. [28] demonstrated that it is necessary to adopt an aerodynamic airfoil for struts, and when two struts per blade are used at intermediate location of 21% and 79% along the blade length, the bending stress distribution along the blade is reduced.

Figure 3 illustrates one rotor stage of F-VAWT. A mechanism consisting of a pivot (screw-nut) is introduced in the middle of the two blades, which is automatically actuated by a small electric motor and a lever in the shape of a parallelogram. The latter is attached to the end of both blades. When turning the screw, the lever moves with the help of the two sliding links and exerts a force on the two blades that allows us to increase the diameter. On the other hand, when the screw rotates in the opposite direction, the diameter will be reduced. This new mechanism aims to change the diameter of the wind turbine automatically according to the wind speed, and it solves the problem of starting a large VAWT rotor.

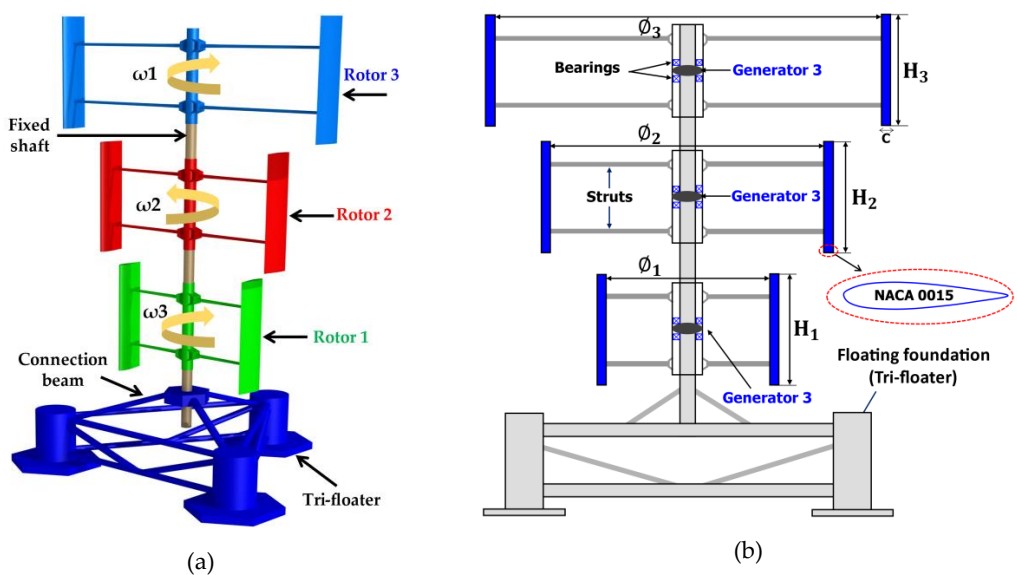

(a)                                           (b)

**Figure 2.** (**a**) 3D design of Floating Darrieus-type wind turbine with three-stage rotors (**b**) Schematic diagram details.

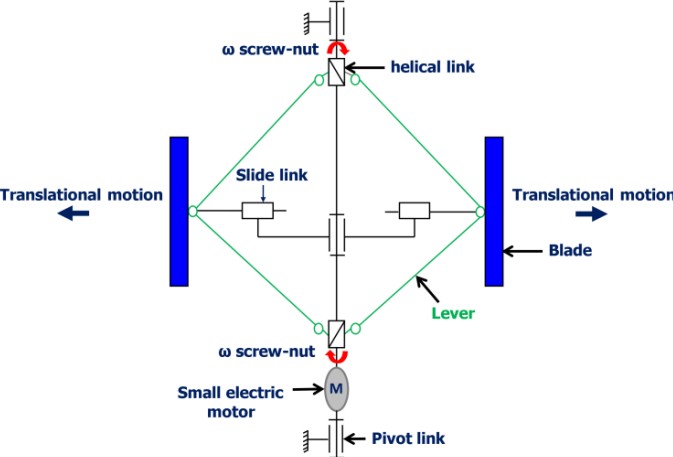

**Figure 3.** Kinematic scheme of the diameter variation mechanism in one rotor stage

## 3. General Mathematical Expressions for Aerodynamic Analysis of SB-VAWT

The general mathematical expressions at a specific location of the blade and the input parameters used for the aerodynamic simulation are described below [25,29,30].

### 3.1. Local Relative Velocity Variation

The wind flow passes over the turbine blades which consist of airfoil cross-section profile to generate useful torque and power. As in case of any airfoil, the chordal and normal velocity components ($V_c$ and $V_n$) can be obtained as follows:

$$V_c = R\omega + V_a \cos\theta \tag{1}$$

$$V_n = V_a \sin\theta \tag{2}$$

where $V_a$ is the axial flow velocity (i.e., induced velocity) through the rotor. A relative velocity component $V_R$ can be obtained from the cordial velocity component and the normal velocity component as can be seen from Figures 4 and 5.

$$V_R = \sqrt{(V_n)^2 + (V_c)^2} = \sqrt{(V_a \sin\theta)^2 + (R\omega + V_a \cos\theta)^2} \tag{3}$$

Tip speed ratio is defined as:

$$TSR = \frac{R\omega}{V_\infty} \tag{4}$$

where R and $\omega$ are the radius and the rotational speed of the rotor, respectively

Dividing Equation (3) by the stream velocity $V_\infty$, the adimensionalized expression of relative velocity is:

$$\frac{V_R}{V_\infty} = \sqrt{((1-a)\sin\theta)^2 + (TSR + (1-a)\cos\theta)^2} \tag{5}$$

where *a* is the axial induction factor.

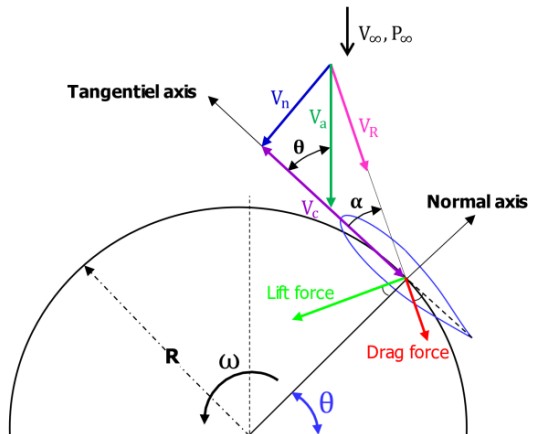

**Figure 4.** Flow velocities diagram of Darrieus VAWT airfoil.

### 3.2. Variation of Local Angle of Attack

A relation between the angle of attack $\alpha$ (incidence angle), the azimuth angle $\theta$, induction factor *a*, and the TSR was defined from the velocity triangle analysis as follows:

$$\tan\alpha = \left(\frac{V_n}{V_c}\right) = \left(\frac{V_a \sin\theta}{R\omega + V_a \cos\theta}\right) \tag{6}$$

By non-dimensionalizing Equation (5) results:

$$\tan\alpha = \left(\frac{\frac{V_a}{V_\infty}\sin\theta}{\frac{R\omega}{V_\infty} + \frac{V_a}{V_\infty}\cos\theta}\right) \tag{7}$$

$$\alpha = \tan^{-1}\left(\frac{(1-a)\sin\theta}{TSR + (1-a)\cos\theta}\right) \tag{8}$$

### 3.3. Normal and Tangential Force Variation

The overall normal and tangential forces are given by:

$$F_n = 0.5 C_n \rho h c V_R^2 \tag{9}$$

$$F_t = 0.5 C_t \rho h c V_R^2 \tag{10}$$

where $\rho = 1.225 \text{ Kg/m}^3$ is the air density, h is blade height, c is the blade chord, $C_n$ the coefficient of the normal force is the difference between the normal components of lift and drag forces

$C_n = C_L\cos\alpha + C_D\sin\alpha$, $C_t$ the coefficient of the tangential force is the difference between the tangential components of lift and drag forces $C_t = C_L\sin\alpha - C_D\cos\alpha$

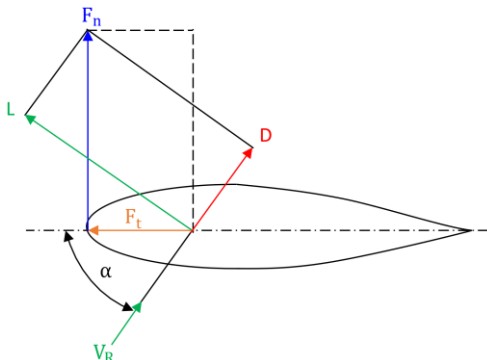

**Figure 5.** Force diagram of a blade airfoil.

### 3.4. Total Torque and Power Output

The normal and tangential forces given by Equations (8) and (9) are for the arbitrary azimuthal position and are regarded as a function of azimuth angle $\theta$. Average tangential force ($F_{ti}$) on one single airfoil at certain $\theta$ is:

$$F_{ti} = \frac{1}{2\pi}\int_0^{2\pi} F_t(\theta)d\theta \tag{11}$$

Considering B blades, the total torque output ($Q_i$) is obtained as follows:

$$Q_i = BF_{ti}R = \frac{BR}{2\pi}\int_0^{2\pi} F_t(\theta)d\theta \tag{12}$$

The total power output ($P_i$) is described by:

$$P_i = Q_i\cdot\omega = \frac{BR\omega}{2\pi}\int_0^{2\pi} F_t(\theta)d\theta \tag{13}$$

### 3.5. Rotor Solidity σ

Rotor solidity is the main parameter to define the geometry of the vertical axis wind turbines, and it is a function of the number of blades B, the chord length c, and the radius of the rotor R. It can be calculated as suggested by Strickland [31]:

$$\sigma = \frac{Bc}{R} \tag{14}$$

### 3.6. Rotor Aspect Ratio AR

Rotor aspect ratio of VAWT defined as the ratio between the blade height and rotor radius:

$$AR = \frac{h}{R} \tag{15}$$

### 3.7. Power Coefficient Cp

Power coefficient $C_P$ is the ratio between the power absorbed by the rotor shaft P and the power available from the air stream flowing through the rotor swept area.

$$C_P = \frac{P}{0.5\rho AV_\infty{}^3} = \frac{Q\omega}{0.5\rho AV_\infty{}^3}\frac{\omega R}{V_\infty} = C_m TSR \tag{16}$$

where P is the mechanical power, ρ is the air density, A is the projected area of the rotor, and $C_m$ is a moment coefficient.

### 3.8. Reynolds Number Re

Reynolds number is used to characterize the flow regime perceived by the blades. It represents the ratio between inertial forces and viscous forces:

$$R_e = \frac{\rho c V_\infty}{\mu} \tag{17}$$

where $\mu = 1.647 \times 10^{-5}$ Kg/m.s is the dynamic viscosity of the air.

## 4. Double Multiple Stream Tube Model

In the literature [25], many aerodynamic models for VAWT have been proposed, such as double multiple stream tube (DMST), multiple stream tube (MST), vortex, cascade, and panel. Each of them has its advantages and disadvantages, and Table 1 gives a comparison of the aerodynamic models according to important criteria such as complexity and computational time. Therefore, it is necessary to find a compromise between the validity of the results, the problem complexity, and the computational effort, depending on the objectives fixed. The model chosen for this work is the DMST developed by Paraschivoiu [32] in 1981. This model combined the MST model with the double actuator disk theory (see Figure 6a) to predict the performance of VAWT. The advantages of this model can be seen through the fact that it is relatively simple to implement and gives better correlation between calculated and experimental results [25]. However, the major drawback of this model is to give over prediction of power for a high solidity turbine, and there appears to be a convergence problem for the same type of turbine, especially in the downstream side and at the higher tip speed ratio [29].

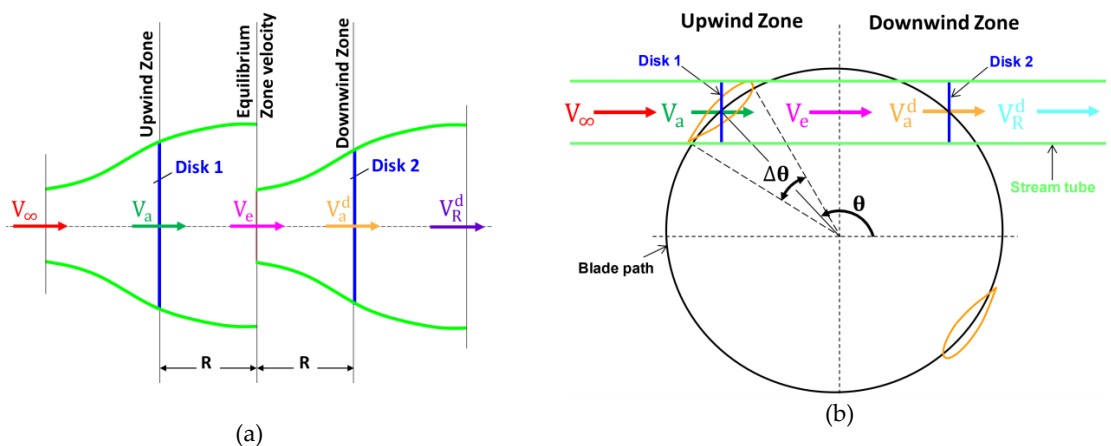

(a)                                    (b)

**Figure 6.** (**a**) Shematic of two-actuator disk in tandem (**b**) Diagrammatic of DMST model.

**Table 1.** Comparison of aerodynamic models [5].

|  | BEM Model | Cascade Model | Vortex Model | Panel Model |
|---|---|---|---|---|
| **Complexity** | Low-Medium | Low-Medium | Medium-High | High |
| **Ease of implementation** | Easy-Medium | Easy-Medium | Medium | Hard |
| **Computational effort** | Low | Low | Medium-High | Medium-High |
| **Restricted to known airfoils** | Yes | Yes | Yes | No |
| **Incorporate unsteady conditions** | Limited | Limited | Yes | Yes |
| **Rotor wake/multiple rotor interactions** | No/No | No/No | Yes/Limited | Yes/Yes |

As shown in Figure 6b, the rotor's area is divided into a set of adjacent parallel stream tubes. The calculation is performed separately for the upwind and downwind half cycles, and the blade

element theories and the momentum conservation are used for calculating the aerodynamic forces acting on the blades, which then makes it possible to calculate the torque and the power generated.

For the upstream half-cycle $\left(\frac{\pi}{2}\right) \le \theta \le \left(\frac{3\pi}{2}\right)$

$$V_R^u = \sqrt{(V_a^u \sin\theta)^2 + (V_a^u \cos\theta + R\omega)^2} \tag{18}$$

$$\alpha^u = \tan^{-1}\left(\frac{(1-a^u)\sin\theta}{TSR + (1-a^u)\cos\theta}\right) \tag{19}$$

For the downwind half-cycle $\left(\frac{3\pi}{2}\right) \le \theta \le \left(\frac{\pi}{2}\right)$

$$V_R^d = \sqrt{\left(V_a^d \sin\theta\right)^2 + \left(V_a^d \cos\theta + R\omega\right)^2} \tag{20}$$

$$\alpha^d = \tan^{-1}\left(\frac{\left(1-a^d\right)\sin\theta}{TSR + (1-a^d)\cos\theta}\right) \tag{21}$$

where, at the downwind side, the induction factor $a^d = \frac{(1-2a)-V_a^d}{(1-2a)}$ and the induced velocity $V_a^d = \left(1-a^d\right)(1-2a)V_\infty$.

Once the new relative wind speed and angle of attack are determined to use the new induction factor, the torque coefficient, thrust coefficient, and power coefficient could be obtained. Equating the forces given by momentum equations to those defined by blade element theory:

$$f_u(1-a) = \pi a \tag{22}$$

where the upwind function $f_u$ is given by:

$$f_u = \frac{Bc}{8\pi R}\int_{\pi/2}^{3\pi/2}\left(C_n\frac{\cos\theta}{|\cos\theta|} - C_t\frac{\sin\theta}{|\sin\theta|}\right)\left(\frac{V_R}{V_a}\right)^2 d\theta \tag{23}$$

Thus, the power coefficient for the upwind half of the turbine is obtained by:

$$C_P = \frac{Bch}{2\pi A}\int_{\pi/2}^{3\pi/2}C_t\left(\frac{V_R}{V_a}\right)^2 d\theta \tag{24}$$

where $A = 2hR$ is the turbine swept area

The downwind part of the rotor is evaluated in the same manner, and finally, by the summation of the power coefficients of the two half-cycles, the total power coefficient of the rotor can be found.

### 4.1. QBlade Simulation Tool

QBlade is used as an open source framework for the design and aerodynamic simulations of the wind turbines. In a recent paper by Nachtane et al. [33] QBlade solver was used to predict the hydrodynamic performance of a new hydrofoil named NTSXX20. QBlade utilizes the blade element momentum (BEM) method for the simulation of HAWTs, and a double multiple stream tube (DMST) model for the VAWTs performance. The XFOIL code is integrated seamlessly into QBlade based on the viscous-inviscid coupled panel method to generate 2D airfoil coordinates for blade design, airfoil lift, drag coefficients, and 360° polar extrapolation for turbine simulations [34]. Figure 7 shows the different modules and types of analysis (shown inside the dotted square) that can be performed by Qblade for aerodynamic simulation.

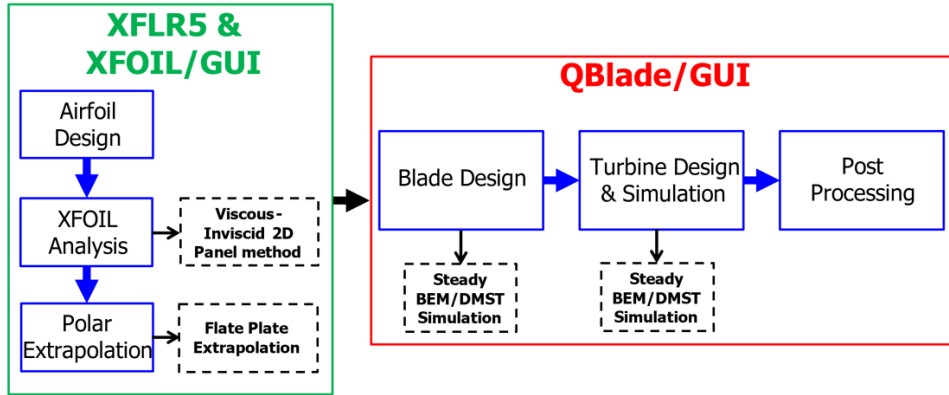

**Figure 7.** Software modules inside QBlade.

## 4.2. QBlade Validation

In order to show the validity of this software, the Qblade was compared with the experimental and computational results of Raciti Castelli et al. [35] with various TSRs. The comparison of the power coefficient versus TSR results from the QBlade with experimental and numerical results are shown in Figure 8. It can be observed that the general trend of the experimental plot was captured by the Qblade where the quantitative differences (presented in Table 2) are found to be the largest for the highest TSR equal to 2.21. The observed deviations may be explained by the uncertainties associated with the experimental data, e.g. boundary conditions, the blade surface roughness, the wake-blade interaction. Also, the effect of blade-spoke connection on the $C_P$ of the turbine is known to be more significant in higher TSRs [36]. As well as Qblade overestimates the $C_P$ at high TSRs due to the dynamic stall effects are not considered, which is difficult to predict because the angle of attack $\alpha$ change rapidly and the turbulence generated. Another source of error is from the extrapolation of the initial data obtained from XFOIL as the airfoil characteristics are required for 360°. Regarding the extrapolation, Montgomerie [37,38] and Viterna [39] are the most widely used methods. The Montgomery method is founded on the assumption as a thin plate, whereas the Viterna method is formulated on the basis of the potential flow theory. Extrapolated values provide a good estimate if the initial limited data are obtained from experiments. However, in this case, the initial data is over predicted by Xfoil which can further extend the high lift/drag ratio error for other angles of attack. Furthermore, the Montgomery and Viterna methods are intended for static airfoils, and do not consider the unsteady effects. The airfoil characteristics calculated from the Xfoil assume that the flow is laminar, but in fact, VAWT airfoils face a turbulent and unstable incoming flow, and it becomes worse in the downstream side, where the blade operates entirely in wake for a high solidity turbine [40].

**Table 2.** Comparison of $C_P$ from experiment, CFD, Qblade and their relative deviation.

| TSR | $C_P$ Experiment | $C_P$ Qblade | $C_P$ CFD | $|\Delta C_P|$ [%] Qblade/Experiment | $|\Delta C_P|$ [%] CFD/Experiment |
|---|---|---|---|---|---|
| 1.69224 | 0.0477033 | 0.0276889 | 0.26065 | 53.09408666 | 138.1186451 |
| 2.00088 | 0.127525 | 0.067902 | 0.412182 | 61.0181807 | 105.4857543 |
| 2.30071 | 0.249668 | 0.141027 | 0.523215 | 55.61422593 | 70.78613451 |
| 2.60494 | 0.310683 | 0.248951 | 0.563603 | 22.06156166 | 57.85749743 |
| 2.90035 | 0.297405 | 0.427411 | 0.554072 | 35.87282841 | 60.28747694 |
| 3.05908 | 0.285123 | 0.453624 | 0.546477 | 45.61805327 | 62.85569986 |
| 3.20899 | 0.267197 | 0.435643 | 0.530408 | 47.93295771 | 66.00033851 |

Table 2 presents percent differences between numerical and experimental results of the power coefficient. The relative deviation between the numerical and experimental results is given by the equation:

$$|\Delta C_P| = \frac{|C_{p1} - C_{p2}|}{\frac{C_{p1} + C_{p2}}{2}} \times 100 \qquad (25)$$

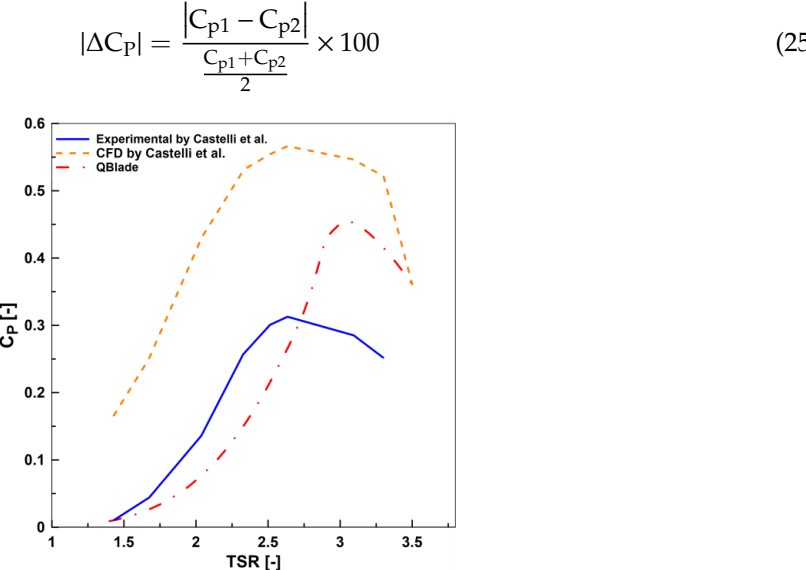

**Figure 8.** Comparison of power coefficient versus experiment and numerical results by Raciti Castelli et al. [35].

### 4.3. Input Parameters Used for the Aerodynamic Simulation

The most common profiles used for commercial Darrieus VAWTs are the symmetrical NACA profiles [10]. The blade airfoil will consist of a symmetrical four-digit NACA 0015 which has a maximum thickness of t/c = 15% and a 0% maximum camber. A study was conducted at the University of Windsor in Canada analyzing the performance of several airfoils for Darrieus VAWT [41]. Of all the airfoils tested, the NACA0015 performed the best performance. Figure 9a gives a comparison between the four aerodynamic profiles in terms of coordinates. To assess the aerodynamic efficiency of each possible VAWT design, the power coefficient Equation (16) will be used. Table 3 outlines the input parameters used for the performance curves for different aerodynamic profiles, and Table 4 gives the physical proprieties of air used for aerodynamic simulation. As shown in Figure 9b, aerodynamic profile NACA 0015 presents the best aerodynamic efficiency compared to other aerodynamic profiles. Figure 10 depicts the lift and drag coefficients of a NACA0015 airfoil at $R_e = 10^6$.

In the following, we have used this profile for the parametric study of the VAWT rotor.

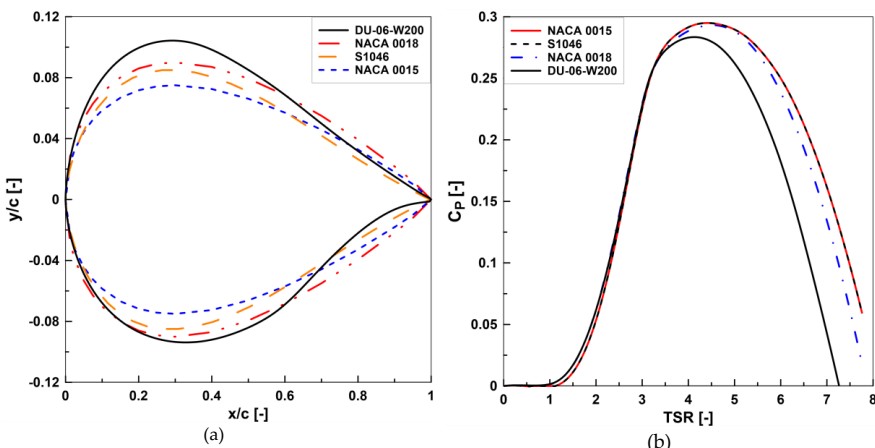

**Figure 9.** (**a**) Comparison between the four aerodynamic profiles in terms of coordinates (**b**) Performance curves for different aerodynamic profiles.

The objective of this study is to carry out a parametric study by the DMST method, in order to show the interest of a stepped rotor with variable radius, and to determine the aerodynamic behavior

of the rotor during the design phase, which then makes it possible to perform a structural analysis of the design.

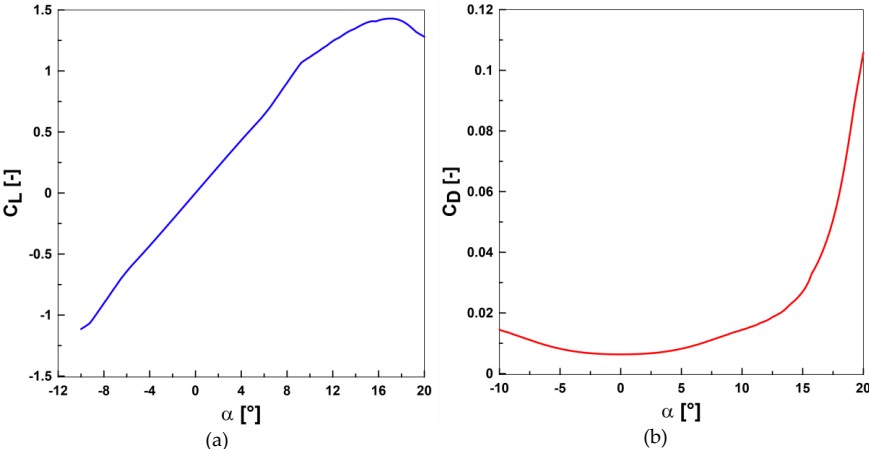

(a) (b)

**Figure 10.** (**a**) Lift coefficient versus angle of attack alpha of NACA 0015 airfoil at $R_e = 10^6$; (**b**) Drag coefficient versus angle of attack alpha of NACA 0015 airfoil at $R_e = 10^6$.

**Table 3.** Influential geometric parameters for aerodynamic simulations.

| Rotor | R [m] | c [m] | h [m] | B | $V_\infty$[m/s] | σ |
|---|---|---|---|---|---|---|
| **1. Aerodynamic airfoils** | | | | | | |
| 1 | 0.7 | 0.09 | 0.45 | 3 | 10 | 0.38 |
| **2. Rotor solidity** | | | | | | |
| 1 | 2 | 0.0667 | | | | 0.1 |
| 2 | 1.5 | 0.1 | 0.45 | 3 | 10 | 0.2 |
| 3 | 1 | 0.1 | | | | 0.3 |
| 4 | 0.7 | 0.0933 | | | | 0.4 |
| **3. Number of blades** | | | | | | |
| 1 | 0.7 | 0.105 | | 2 | | |
| 2 | 1 | 0.1 | 0.45 | 3 | 10 | 0.3 |
| 3 | 1.5 | 0.1125 | | 4 | | |
| 4 | 2 | 0.12 | | 5 | | |
| **4. Rotor radius** | | | | | | |
| 1 | 0.7 | 0.105 | | | | |
| 2 | 1 | 0.15 | 0.45 | 2 | 10 | 0.3 |
| 3 | 1.5 | 0.225 | | | | |
| 4 | 2 | 0.3 | | | | |
| **5. Rotor aspect ratio** | | | | | | |
| 1 | 0.7 | 0.105 | 0.28 | | | |
| 2 | 1 | 0.15 | 0.8 | | | |
| 3 | 1 | 0.15 | 1 | 2 | 10 | 0.3 |
| 4 | 1.5 | 0.225 | 1.8 | | | |
| 5 | 2 | 0.3 | 3.2 | | | |
| **6. Wind speed/Rotor height** | | | | | | |
| 1 | 2.5 | 0.375 | 1 | | 10 | |
| 2 | 3.75 | 0.5625 | 1.5 | 2 | 12 | 0.3 |
| 3 | 5 | 0.75 | 2 | | 14 | |

**Table 4.** Physical properties of air.

| $R_e$ | $\rho_{air}$ [Kg/m$^3$] | $\mu_{air}$ [Kg/m.s] |
|---|---|---|
| $10^6$ | 1.225 | $1.647 \cdot 10^{-5}$ |

## 5. Results and Discussion

### 5.1. Rotor Solidity

The influence of altering the turbine solidity on its aerodynamic performance has been studied using the DMST model as shown in Figure 11. The solidity σ was increased from 0.1 to 0.4 by decreasing the turbine radius while the blades swept area was kept constant (see Table 3). The DMST model has been run many times for H-rotor with three NACA 0015 blades and Reynolds number $10^6$. It can be observed for σ = 0.3 that the $C_P$ grows rapidly compared to the other solidity. This is illustrated clearly in Figure 11 where the $C_P$ peak (0.342) expected at a low tip speed ratio (TSR = 3.51).

As the turbine solidity decreases (σ = 0.1), the $C_P$ drops (0.183) at high TSR (6.51). Therefore, from a structural design standpoint, a low solidity turbine will experience greater centrifugal forces on the blades and supporting arms because of its higher TSR operating conditions [25,29]. While these forces could be contained with the use of modern composite materials. It can be concluded that the optimum solidity (σ = 0.3) was chosen for this VAWT design as it has achieved a higher aerodynamic efficiency, the same optimum solidity has been found in other works [42,43].

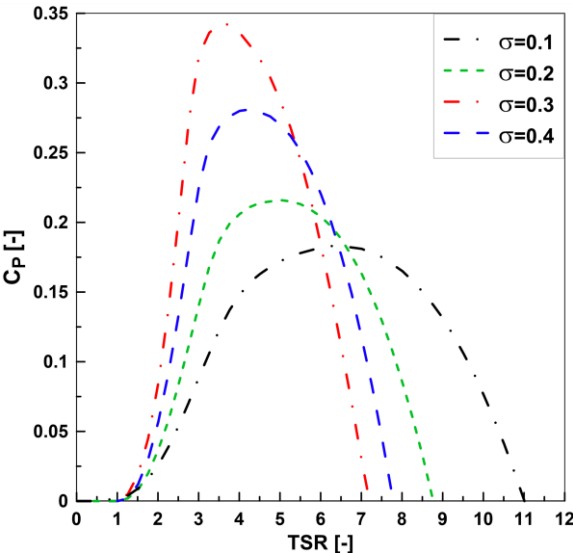

**Figure 11.** Performance curve for different solidity.

### 5.2. Number of Blades

Increasing the number of blades is to increase the number of wakes and accentuate the interactions of the blades with the wakes. Nevertheless, this has the advantage of smoothing the instantaneous torque produced by the rotor and decreasing the radial component of the aerodynamic forces [44]. Furthermore, for a specific turbine solidity, the blade becomes leaner and subsequently has less resistance to bending when using more than two or three blades, which requires more struts, and this creates more parasitic drag. As a result, the aerodynamic performances of turbines decrease significantly, which is undesirable from a structural perspective. Therefore, the choice of the number of blades is a compromise between aerodynamic efficiency, the blade stiffness, and cost considerations [45]. The impact of increasing the number of blades on the turbine aerodynamic efficiency is given in Figure 12, the solidity was kept constant (Table 3). It is found that the most identifiable change in rotor

efficiency is from two to three blades. We have chosen a two-bladed, as it allows minimizing both the rotor weight, cost and the problem of starting the turbine. Also, with the two-bladed design, we can integrate the diameter variation mechanism (Figure 3) that allows varying the rotor radius according to the wind velocity available.

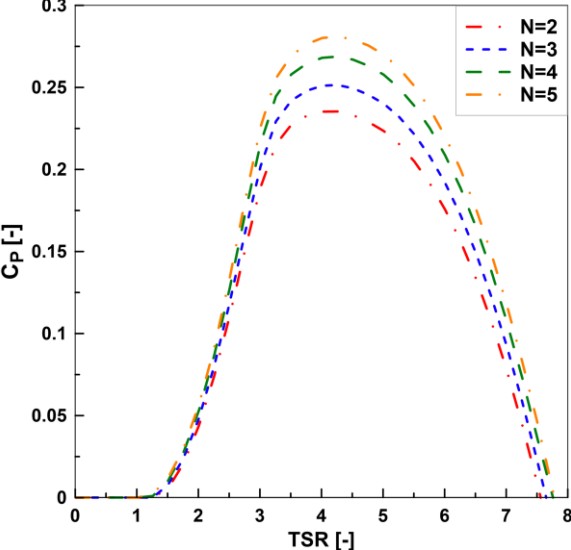

**Figure 12.** Effect of the number of blades on the turbine efficiency.

### 5.3. Rotor Radius

In the literature, few research studies explore the influence of the rotor radius on the performance of VAWT. The variation in Darrieus rotor radius in solo Darrieus and a combined Darrieus–Savonius wind turbine has an important effect on the turbine performance. Figure 13a shows the variation of the power coefficient $C_P$ with tip speed ratio for four different radii, although the solidity and number of blades were kept constant (Table 3). The $C_p$ attains its highest value of the tip speed ratio TSR = 4.96 for radius equal to 0.7 m and it decreases significantly with the increase of the radius. We find the same result found by Sahim et al. [46], and the numerical results show the interest of a variable radius rotor, whereby when we increase the rotor radius, obviously, the $C_P$ decreases. In contrast, the power generated by the rotor increases when the rotor radius increases, as shown in Figure 13b.

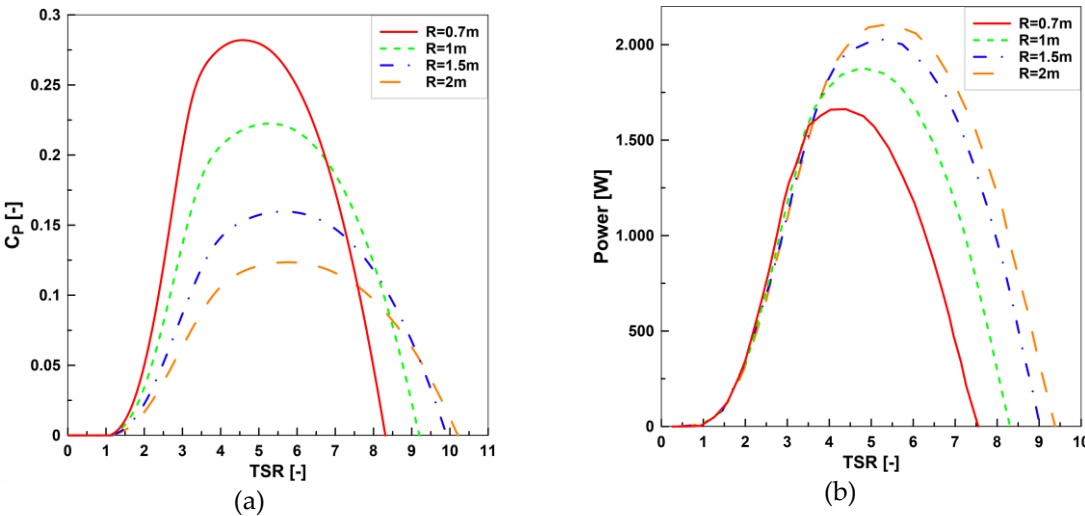

**Figure 13.** (**a**) Power coefficient Cp versus TSR for different radius; (**b**) Mechanical Power versus TSR for different radius.

### 5.4. Rotor Aspect Ratio

As shown in Table 3, five H-Darrieus rotors with different radius, chord length, and height are considered to investigate the effect of altering the aspect ratio on the aerodynamic performance of the rotor. Figure 14 represents the obtained results of increasing the blade aspect ratio on the rotor's power coefficient. It is clear that from aerodynamic point the peak of the $C_p$ increases with high aspect ratio, and the maximum power coefficient value is 0.469 at TSR = 4.01. However, from a structural point, a rotor with high aspect ratio is subjected to high bending moments, so during the design phase, it should consider the aerodynamic performance aspects and the structural resistance aspects should be considered. Indeed, floating Darrieus-type wind turbines with three-stage rotors contain these two aspects because the rotor is subdivided into three stages with a different radius and height. This means a different AR which will allow us to vary the output performance. Therefore, this subdivision will also solve the problem of starting a single large turbine.

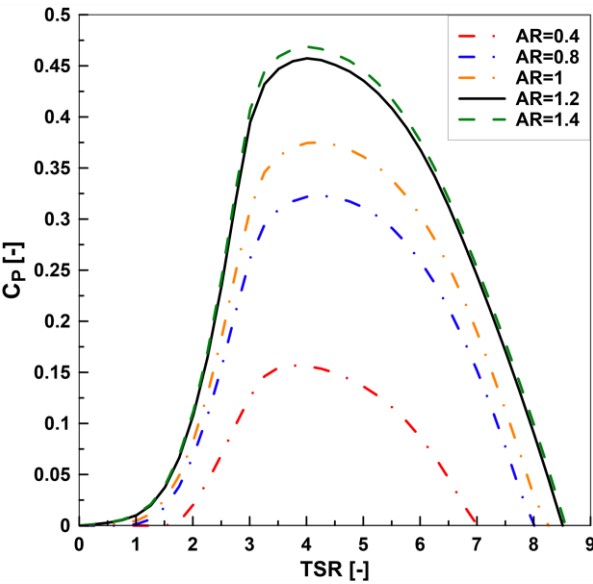

**Figure 14.** Effect of increasing the AR on the rotor's power coefficient.

### 5.5. Wind Speed with Rotor Height

The rotor's power efficiency was evaluated for a range of free stream velocity from 10 m/s to 15 m/s (the rotor high varied from 1 m to 1.5 m, respectively) at different tip speed ratio ranging from 1.5 to 7, with the influential geometric parameters are given in Table 3. Considering Figure 15, which plots TSR as a function of the power output, it indicates that the power peak will augment without altering its optimal TSR when the free stream velocity increases. In addition, for large values of free stream velocity and rotor height, the power increases considerably compared to the low values of free stream velocity and rotor height. Hence, this showed the influence of the variation of free stream velocity (height rotor, respectively) on the power generated by the rotor.

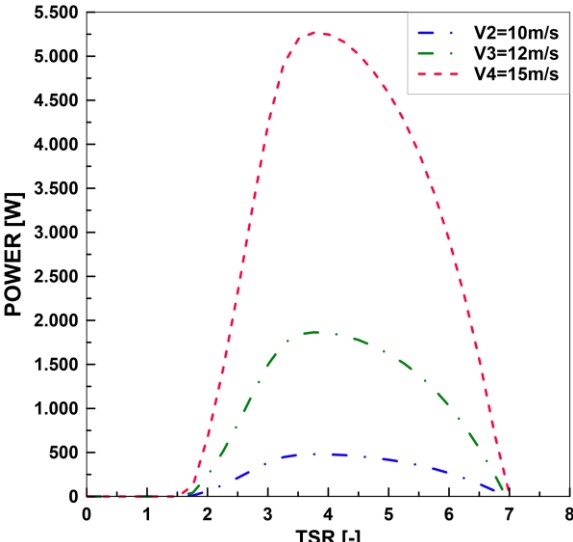

**Figure 15.** Effect of increasing wind velocity and height on the rotor's power coefficient.

## 6. Conclusions

This paper presents a new design of F-VAWT with three-stage rotors. Three electrical generators can be used with different powers depending on the rotor radius, wind speed, and the height where each rotor is located. In fact, this configuration of three stage rotors helps to optimize power and solve the problem of turbine self-starting. The result of the performance curve of Qblade software was validated by comparison with experimental and numerical results, and it shows a good prediction of the rotor performance. Moreover, the aerodynamic profile NACA 0015 is employed for aerodynamic simulations, because this profile gives the best aerodynamic efficiency compared to other aerodynamic profiles, as well as being the most common profiles used for commercial Darrieus VAWTs. An investigation of several influential geometric parameters of VAWT has been taken into account, including solidity, number of blades, rotor radius, aspect ratio, wind velocity, and rotor height. Thus, numerical results obtained by the aerodynamic simulations identify a low solidity turbine ($\sigma = 0.3$), offering the best aerodynamic performance. A two-blade design is recommended to minimize the rotor weight, reduce the problem of starting the turbine, and lower the cost. In addition, we can integrate a mechanism that allows varying the rotor radius according to the wind speed available by using a two-bladed design. These findings also indicate the interest of a variable AR and wind velocity, as when these parameters increased, the aerodynamic performance will be improved as well.

**Author Contributions:** Conceptualization and writing—original draft preparation, M.A.D. and A.R.; methodology, O.B.; software and validation, M.A.D.; investigation, formal analysis, writing—review and editing, visualization, E.R. All authors have read and agreed to the published version of the manuscript.

**Funding:** This research received no external funding.

**Acknowledgments:** The work of the third author was supported by the project "Excellence, performance and competitiveness in the Research, Development and Innovation activities at "Dunarea de Jos" University of Galati", acronym "EXPERT", financed by the Romanian Ministry of Research and Innovation in the framework of Programme 1—Development of the national research and development system, Sub-programme 1.2—Institutional Performance —Projects for financing excellence in Research, Development and Innovation, Contract no. 14PFE/17.10.2018.

**Conflicts of Interest:** The authors declare no conflict of interest.

## Nomenclature

| | |
|---|---|
| A | turbine swept area, m$^2$ (A = 2hR) |
| $a_u$, $a_d$ | upwind and downwind induction factors |
| B | number of blades |
| c | blade chord length, m |
| $C_D$ | drag coefficient |
| $C_L$ | lift coefficient |
| $C_n$ | normal coefficient |
| $C_m$ | moment coefficient |
| $C_t$ | tangential coefficient |
| $C_P$ | power coefficient |
| D | blade drag force, N |
| $f_u$, $f_d$ | upwind and downwind functions |
| $F_n$, $F_t$ | normal and tangential force, N |
| $F_{ti}$ | average tangential force, N |
| h | height of the blade/turbine, m |
| L | blade lift force, N |
| $P_i$ | total power output |
| $Q_i$ | total torque output |
| R | radius of the wind turbine, m |
| $R_e$ | local Reynold number |
| $V_a$ | upstream induced velocity, m/s |
| $V_a^u$, $V_a^d$ | upstream and downstream induced velocity, m/s |
| $V_c$ | chordal velocity component, m/s |
| $V_n$ | normal velocity component, m/s |
| $V_R^u$ | relative velocity upstream, m/s |
| $V_R^d$ | relative velocity downstream, m/s |
| $V_\infty$ | free-stream velocity, m/s |
| $\alpha$ | blade angle of attack |
| $\alpha^u$, $\alpha^d$ | upstream and downstream angle of attack |
| $\Delta\theta$ | azimuth angle increment, correspond to one stream tube |
| $\theta$ | azimuth angle |
| TSR | tip speed ratio |
| $\rho$ | density of the air (kg/m$^3$) |
| $\mu$ | dynamic viscosity of the air (kg/m.s) |
| $\sigma$ | solidity |
| $\omega$ | angular velocity of rotor, rad/s |

## Acronyms

| | |
|---|---|
| DMST | double multiple stream tube |
| F-VAWTs | floating vertical axis wind turbines |
| F-HAWTs | floating horizontal axis wind turbines |
| HAWT | horizontal axis wind turbine |
| MST | multiple stream tube |
| SKWID | Savonius keel wind turbine Darrieus |
| SB-VAWT | straight-bladed vertical axis wind turbine |
| PMSMs | permanent-magnet synchronous motors |

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
