# Peer review of "Aerodynamic Simulations for Floating Darrieus-Type Wind Turbines with Three-Stage Rotors"

_inventions, doi:10.3390/inventions5020018_

Round 1

Reviewer 1 Report

Dear Authors, I really like the idea of a three-stage rotor. However, I have some concerns listed below:

Although the concept of a three-stage rotor F-VAWT is novel, the paper does not concentrate on possible advantages of the concept. By means of the presented model different scenarios, e.g. with reported wind velocities at different altitudes, could have been analyzed to show performance advantages against single stage F-VAWT. This way readers might be convinced of the advantages of this concept although it might bear higher costs in the manufacturing. However, the basis for the aforementioned analysis is a well validated model.

The introduction gives a good overview of the field. Figure 1 supports well the introduction. However, the descriptions inside the Figure 1 seem to be shifted. Please adjust them.

Line 96: “Section II”, S is capital letter.

Line 112: Can you describe in more detail how the selfstarting is solved by the presented setup

Figure 2: Also here the text inside the figure seems to be shifted and has a few errors. Please revise. I would recommend to place omega symbol at the rotation axis. Since the rotation can be independent it would be intuitive to name them omega_1, omega_2, and omega_3.

Line 190: can you please refer with more detail to the reasons why you have selected the DMST. What are the advantages precisely for your work?

Figure 6: revise text positions. In b), the V plus subscripts and angles are not displayed properly.

Line 232: Do the numerical DMST results also represent a NACA 0021 profile as the experiment? If so, the validity of the presented model is not given qualitatively nor quantitatively. At low TSR e.g. 2 the power coefficient is 50% lower as in the experiment. Maximum are neither at the same TSR nor have a similar value. If both profiles are not NACA 0021, an appropriate comparison of the same profiles should be presented to show the validity of the model. Please take into account that this is crucial for the credibility of the model. At least the qualitative trend of power coefficient with changing solidity should be validated, so that following analysis can be strengthened.

Line 300: “Cp”, p as subscript.

Author Response

ANSWERS TO THE REVIEWERS' COMMENTS 

The specific corrections operated according to the suggestions of the reviewers are given next together with detailed explanations.

Respected Editor and Reviewers!

We greatly appreciate the valuable comments made by the reviewers. These comments are very valuable and helpful for revising and improving our paper, as well as for the guidance of our future research.

After carefully studying the comments, we have made corresponding changes, which are highlighted with ''track changes'' in the revised manuscript. Next are provided the answers and revisions we have made in responding to the reviewer’ comments on an item-by-item basis.

-----------------------------------------------------------------------------------------------------------------

Reviewer # 1:

  1. Dear Authors, I really like the idea of a three-stage rotor.

Thank you for these comments and for the appreciation of our work.

  1. However, I have some concerns listed below:

Although the concept of a three-stage rotor F-VAWT is novel, the paper does not concentrate on possible advantages of the concept. By means of the presented model different scenarios, e.g. with reported wind velocities at different altitudes, could have been analyzed to show performance advantages against single stage F-VAWT. This way readers might be convinced of the advantages of this concept although it might bear higher costs in the manufacturing. However, the basis for the aforementioned analysis is a well validated model.

Thank you for this valuable suggestion made by the reviewer. This comment is now considered in revised paper and following this observation we have added the subsections (3.4 rotor aspect ratio, and 5.5 wind speed with rotor height) to results section (lines 334-358) and copied below:

5.4 Rotor aspect ratio

As shown in Table 3.5, five H-Darrieus rotors with different radius, chord length, and high are considered to investigate the effect of altering the aspect ratio on the aerodynamic performance of the rotor. Figure 14 represents the obtained results of increasing the blade aspect ratio on the rotor's power coefficient. It is clear that from aerodynamic point the peak of the  increases with high aspect ratio, and the maximum power coefficient value is 0.469 at TSR = 4.01. However, from a structural point, a rotor with high aspect ratio is subjected to high bending moments, so during the design phase, it should consider the aerodynamic performance aspects and the structural resistance aspects should be considered. Indeed, Floating Darrieus-type wind turbines with three-stage rotors contain these two aspects because the rotor is subdivided into three stages with a different radius and height. This means a different AR which will allow us to vary the output performance. Therefore, this subdivision will also solve the problem of starting single large turbine.

Table 3: Influential geometric parameters for aerodynamic simulations.

Rotor

R [m]

c [m]

h [m]

B

  [m/s]

?

1. Aerodynamic airfoils

1

0.7

0.09

0.45

3

10

0.38

2. Rotor solidity

1

2

0.0667

0.45

3

10

0.1

2

1.5

0.1

0.2

3

1

0.1

0.3

4

0.7

0.0933

0.4

3. Number of blades

1

0.7

0.105

0.45

2

10

0.3

2

1

0.1

3

3

1.5

0.1125

4

4

2

0.12

5

4. Rotor radius

1

0.7

0.105

0.45

2

10

0.3

2

1

0.15

3

1.5

0.225

4

2

0.3

5. Rotor aspect ratio

1

0.7

0.105

0.28

2

10

0.3

2

1

0.15

0.8

3

1

0.15

1

4

1.5

0.225

1.8

5

2

0.3

3.2

6. Wind speed/Rotor height

1

2.5

0.375

1

2

10

0.3

2

3.75

0.5625

1.5

12

3

5

0.75

2

14

Figure 14. Effect of increasing the AR on the rotor's power coefficient

5.5 Wind speed with rotor height

Figure 15. Effect of increasing wind velocity and height on the rotor's power coefficient.

The rotor’s power efficiency was evaluated for a range of free stream velocity from 10 m/s to 15 m/s (the rotor high varied from 1 m to 1.5 m, respectively) at different tip speed ratio ranging from 1.5 to 7, with the influential geometric parameters are given in Table 4.6. Considering Figure 15, which plots TSR as a function of the power output, it indicates that the power peak will augment without altering its optimal TSR when the free stream velocity increases. In addition, for large values of free stream velocity and rotor height the power increases considerably compared to the low values of free stream velocity and rotor height. Hence, this showed the influence of the variation of free stream velocity (height rotor, respectively) on the power generated by the rotor.

  1. The introduction gives a good overview of the field. Figure 1 supports well the introduction. However, the descriptions inside the Figure 1 seem to be shifted. Please adjust them.

Thank you for this suggestion. We have adjusted the description inside Figure 1. Thus, the revised manuscript was included with Figure 1 in introduction (lines 67-69)

Figure 1. Types of Darrieurs VAWT.

  1. Line 96: “Section II”, S is capital letter.

Following this suggestion of the reviewer, we have corrected the capitalization error of the letter S (line 96).

  1. Line 112: Can you describe in more detail how the selfstarting is solved by the presented setup.

The proposed design of floating Darrieus-type wind turbines with three-stage rotors solved the problem of starting large turbine, because in our design the rotor is subdivided into three stages with a different radius and height, which means a different AR which will allow us to vary the output performance, also by using two-bladed design, we can integrate a mechanism that allows varying the rotor radius according to the wind velocity available.

  1. Figure 2: Also, here the text inside the figure seems to be shifted and has a few errors. Please revise. I would recommend to place omega symbol at the rotation axis. Since the rotation can be independent it would be intuitive to name them omega_1, omega_2, and omega_3.

Thank you very much for reviwer’s guidance, we worked on the 3D design floating Darrieus-type wind turbines with three-stage rotors (Figure 2a) and we improved the schematic diagram (Figure 2b) according to your honorable suggestion (lines 99-100).

(a)

(b)

Figure 2. (a) 3D design of Floating Darrieus-type wind turbine with three-stage (b) Schematic diagram details

  1. Line 190: can you please refer with more detail to the reasons why you have selected the DMST. What are the advantages precisely for your work?

To incorporate the reviewers’ comment here, as you know that the 3D CFD method takes a lot of time in terms of calculation especially for large structures (as in our case), this is why we chose the DMST method because we are in the first phase of the design, and the DMST gives acceptable results.

we have indicated more clearly the advantage and also disadvantages of DMST as follow (lines 187-201):

In the literature [25], many aerodynamic models for VAWT have been proposed, such as Double Multiple Stream Tube (DMST), Multiple Stream Tube (MST), Vortex, Cascade, Panel etc. Each of them has its advantages and disadvantages, Table 1 gives a comparison of the aerodynamic models according to important criteria such as: complexity, computational time etc. Therefore, it is necessary to find a compromise between the validity of the results, the problem complexity, and the computational effort, depending on the objectives fixed. The model chosen for this work is the DMST developed by Paraschivoiu in 1981 [32]. This model combined the MST model with the double actuator disk theory (see Figure 6a) to predict the performance of VAWT. The advantages of this model can be seen through the fact that it is relatively simple to implement and gives better correlation between calculated and experimental results [25]. However, The major drawback of this model is to give over prediction of power for a high solidity turbine and there appears to be a convergence problem for the same type of turbine, especially in the downstream side and at the higher tip speed ratio [29].

Table 1: Comparison of aerodynamic models [5]

BEM model

Cascade model

Vortex model

Panel model

Complexity

Low-Medium

Low-Medium

Medium-High

High

Ease of implementation

Easy-Medium

Easy-Medium

Medium

Hard

Computational effort

Low

Low

Medium-High

Medium-High

Restricted to known airfoils

Yes

Yes

Yes

No

Incorporate unsteady conditions

Limited

Limited

Yes

Yes

Rotor-wake/multiple rotor interactions

No/No

No/No

Yes/Limited

Yes/Yes

As shown in Figure 6b, the rotor's area is divided into a set of adjacent parallel stream tubes. The calculation is performed separately for the upwind and downwind half cycles, the blade element theories and the momentum conservation are used for calculating the aerodynamic forces acting on the blades, which then makes it possible to calculate the torque and the power generated.

  1. Figure 6: revise text positions. In b), the V plus subscripts and angles are not displayed properly.

(a)

(b)

Figure 6. (a) Shematic of two-actuator disk in tandem (b) Diagrammatic of DMST model.

  1. Line 232: Do the numerical DMST results also represent a NACA 0021 profile as the experiment? If so, the validity of the presented model is not given qualitatively nor quantitatively. At low TSR e.g. 2 the power coefficient is 50% lower as in the experiment. Maximum are neither at the same TSR nor have a similar value. If both profiles are not NACA 0021, an appropriate comparison of the same profiles should be presented to show the validity of the model. Please take into account that this is crucial for the credibility of the model. At least the qualitative trend of power coefficient with changing solidity should be validated, so that following analysis can be strengthened.

Thank you very much for reviewer’s guidance. In the literature, we only have this information. It is for this reason that the comparison between experimental and numerical data was made only for NACA0021. We validated the Qblade software with other experimental and numerical results to show the validity of the software. Once we showed this, a parametric study was performed by changing the solidity, number of blades, radius...).

This comment is now considered in the revised manuscript and we have added the section of Software validation in the revised manuscript as follow (lines 221– 260):

Software validation

Figure 7. Software modules inside QBlade.

QBlade is used as an open source framework for the design and aerodynamic simulations of the wind turbines. In a recent paper by Nachtane et al. [34] QBlade solver was used to predict the hydrodynamic performance of a new hydrofoil named NTSXX20. QBlade utilizes the Blade Element Momentum (BEM) method for the simulation of HAWTs, and a Double Multiple Stream tube (DMST) model for the VAWTs performance. The XFOIL code is integrated seamlessly into QBlade based on the viscous-inviscid coupled panel method to generate 2D airfoil coordinates for blade design, airfoil lift, drag coefficients and 360° polar extrapolation for turbine simulations [35]; Figure 7 shows the different modules and types of analysis (shown inside the dotted square) that can be performed by Qblade for aerodynamic simulation.

In order to show the validity of this software, the Qblade was compared with the experimental and computational results of Raciti Castelli et al. [36] with various TSRs. The comparison of the power coefficient versus TSR results from the QBlade with experimental and numerical results are shown in Figure 8. It can be observed that the general trend of the experimental plot was captured by the Qblade where the quantitative differences (presented in Table 2) are found to be the largest for the highest TSR equal to 2.21. The observed deviations may be explained by the uncertainties associated with the experimental data, e.g. boundary conditions, the blade surface roughness, the wake-blade interaction. Also, the effect of blade-spoke connection on the  of the turbine is known to be more significant in higher TSRs [37]. As well as Qblade overestimates the  at high TSRs due to the dynamic stall effects are not considered, which is difficult to predict because the angle of attack change rapidly and the turbulence generated. Another source of error is from the extrapolation of the initial data obtained from XFOIL as the airfoil characteristics are required for 360˚. Regarding the extrapolation, Montgomerie [38, 39] and Viterna [40] are the most widely used methods. The Montgomery method is founded on the assumption as a thin plate, whereas the Viterna method is formulated on the basis of the potential flow theory. Extrapolated values provide a good estimate if the initial limited data are obtained from experiments. However, in this the case the initial data is an over predicted by Xfoil which can further extend the high lift/drag ratio error for other angle of attack. Furthermore, the Montgomery and Viterna methods are intended for static airfoils, which did not consider the unsteady effects. The airfoil characteristics calculated from the Xfoil assume that the flow is laminar, but in fact, VAWTs airfoils face a turbulent and unstable incoming flow, and it becomes worse in the downstream side, where the blade operates entirely in wake for a high solidity turbine [41].

Figure 8. Comparison of power coefficient versus experiment and numerical results by Raciti Castelli et al. [36].

Table 2 present percent differences between numerical and experimental results of the power coefficient. The relative deviation between the numerical and experimental results is given by the equation:

(25)

Table 2: Comparison of  from experiment, CFD, Qblade and their relative deviation

TSR

Experiment

Qblade

 CFD

 [%] Qblade/Experiment

 [%] CFD/Experiment

1,69224

0,0477033

0,0276889

0,26065

53,09408666

138,1186451

2,00088

0,127525

0,067902

0,412182

61,0181807

105,4857543

2,30071

0,249668

0,141027

0,523215

55,61422593

70,78613451

2,60494

0,310683

0,248951

0,563603

22,06156166

57,85749743

2,90035

0,297405

0,427411

0,554072

35,87282841

60,28747694

3,05908

0,285123

0,453624

0,546477

45,61805327

62,85569986

3,20899

0,267197

0,435643

0,530408

47,93295771

66,00033851

Thank you for this comment.

  1. Line 300: “Cp”, p as subscript.

Following this suggestion of the reviewer, we have added the Cp in the subscript list (line 388).

Reviewer 2 Report

Summary

The authors propose a new type of vertical, Darrieus-type wind turbine which - other than conventional turbines of this design - has three stages. The fundamentals of 2D airfoil analyses as well as models for the analysis of wind turbines are presented. A design approach is introduced and applied in a parameter study of stage parameters.

Paper Organization

The paper is organized in six chapters.

  • Chapter 1: An overview on the development of different types of wind turbines is given.
  • Chapter 2: The three-stage concept by the authors is discussed.
  • Chapter 3: The methodology of computing the power output in a 2D model is discussed.
  • Chapter 4: The workflow of an aerodynamic parameter study is discussed.
  • Chapter 5: The results of the parameter study are presented.
  • Chapter 6: Concluding remarks

Content

Text - Scope and overall content of the paper

The major flaw in the content of the paper is that the conducted analysis does not address the problems of the proposed invention. The authors propose to expand a single stage turbine to three stages. The analysis, however, is only conducted on a single stage turbine (at least it apears so from what is presented) because only 2D aerodynamic effects are examined and spanwise interactions between the stages are not part of the analysis. All the varied parameters (blade profiles, TSR, N, solidity, R) characterize a single stage but not a three-stage turbine. The analysis is therefore not sufficient to assess the benefit of the new design but it is rather outlines a performance analysis approach of conventional single-stage turbines.

The authors have two options to resolve this problem:

  1. Redo the analysis, taking stage interaction into account
    In order to provide sufficient novelty for a scientific publication, the authors must include the following points in their analysis:
    ⦁ Include the spanwise interaction of stages in their model
    ⦁ Conduct an analysis of model parameters of the full, three-stage model, not only of single stage parameters
    ⦁ Make a recommendation on how to properly design a three-stage turbine aerodynamically
  2. Leave out the expansion to three stages
    Since these changes would essentially require conducting a completely new analysis and the re-writing of the entire paper, the authors might consider changing the scope of their paper and removing the connection to three-stage rotors. In this case, they might leave out Section 2 and focus solely on their design approach for single stage turbines. This would include changing the title of the paper and adjusting the content of the paper accordingly.

Text - Description of design approach

In both cases, the description of the design approach, i.e. the discussion of Fig. 7, needs to be reworked. From Fig. 7 it is not clear, what the respective components of QBlade do. The tool XFoil is tyically used for 2D airfoil analysis. It is unclear, how the results of Xfoil are used for a subsequent blade design. The used software modules must be described in much more detail and connected to the analyses of Section 3 and 4.

Text- Other

  • Provide a source for the 59.3% power of Betz' limit (p. 2)
  • Fig. 6a): Why is this Figure shown? It is uncelar which part of the text it refers to.
  • "Advantages and disadvantages" of aerodynamic models on p. 7 must be stated more clearly
  • Fig. 9: What are these parameters based on? What is the stage solidity? Which stage is analyzed in the following?
  • TSR greater than 3.5 are mentioned on p. 9 but not shown in Fig. 8.
    Why are the results by Li et al. (peak of Cp decreased with increase of N) not reproduced in Fig. 11?

Figures

  • One of the blades in Fig. 6 b) is illustrated upside-down

Equations

  • The density is missing in the definition of the Reynolds number (Eqn. 15)
  • In the definitions of forces acting on the blade (Eqns. 8, 9, 10), the velocity V must be V^2

Style

Figures

  • The overall quality of the Figures is very low. Figures 1, 2, 4, 6 must be reworked because the text in the PDF is either not legible, too small or symbols are not printed or printed on top of each other in the PDF.

Equations

  • Equations 10, 17, 18 appear twice in the document.
  • Equations 12, 13 are not legible in the document.
  • TSR and a must be defined before Eqn. 4.
  • Typesetting (indices and exponents) in the presentation of equations and units should be corrected

Abbreviations

  • Some abbreviations are either not defined (e.g. FAWT) or defined twice (e.g. DMST).

Text

  • At multiple occasions in the text, lists are concluded with "...". This should be avoided.
  • The keyword "section" is not capitalized consistently.

Author Response

ANSWERS TO THE REVIEWERS' COMMENTS 

The specific corrections operated according to the suggestions of the reviewers are given next together with detailed explanations.

Respected Editor and Reviewers!

We greatly appreciate the valuable comments made by the reviewers. These comments are very valuable and helpful for revising and improving our paper, as well as for the guidance of our future research.

After carefully studying the comments, we have made corresponding changes, which are highlighted with ''track changes'' in the revised manuscript. Next are provided the answers and revisions we have made in responding to the reviewer’s comments on an item-by-item basis.

-----------------------------------------------------------------------------------------------------------------

Reviewer # 2:

  1. The authors conducted an interesting study to monitor the performance of an H-Darrieus Vertical Axis Wind Turbine using DMS technic.

Thank you for these comments and for the appreciation of our work.

  1. Summary

The authors propose a new type of vertical, Darrieus-type wind turbine which - other than conventional turbines of this design - has three stages. The fundamentals of 2D airfoil analyses as well as models for the analysis of wind turbines are presented. A design approach is introduced and applied in a parameter study of stage parameters.

  1. Paper Organization

The paper is organized in six chapters.

Chapter 1: An overview on the development of different types of wind turbines is given.

Chapter 2: The three-stage concept by the authors is discussed.

Chapter 3: The methodology of computing the power output in a 2D model is discussed.

Chapter 4: The workflow of an aerodynamic parameter study is discussed.

Chapter 5: The results of the parameter study are presented.

Chapter 6: Concluding remarks

  1. Content

4.1 Text - Scope and overall content of the paper

The major flaw in the content of the paper is that the conducted analysis does not address the problems of the proposed invention. The authors propose to expand a single stage turbine to three stages. The analysis, however, is only conducted on a single stage turbine (at least it apears so from what is presented) because only 2D aerodynamic effects are examined and spanwise interactions between the stages are not part of the analysis. All the varied parameters (blade profiles, TSR, N, solidity, R) characterize a single stage but not a three-stage turbine. The analysis is therefore not sufficient to assess the benefit of the new design, but it is rather outlines a performance analysis approach of conventional single-stage turbines.

The authors have two options to resolve this problem:

  • Redo the analysis, taking stage interaction into account

In order to provide sufficient novelty for a scientific publication, the authors must include the following points in their analysis:
Include the spanwise interaction of stages in their model
Conduct an analysis of model parameters of the full, three-stage model, not only of single stage parameters
Make a recommendation on how to properly design a three-stage turbine aerodynamically

Thank you very much for this valuable suggestion made by the reviewer, I totally agree with you, however, the problem that the interaction with this configuration (three stepped rotors ) can be studied only with the 3D CFD method, it is impossible to study the interactions between these three rotors with a 2D method (vortex, panel, 2D CFD), because they are not in the same plane. Furthermore, as you know that the 3D CFD method takes a lot of time in terms of calculation especially for large structures (as in our case), this is why we chose the DMST method because we are in the first phase of the design. Hence the DMST is an open source software and gives acceptable results.

  • Leave out the expansion to three stages

Since these changes would essentially require conducting a completely new analysis and the re-writing of the entire paper, the authors might consider changing the scope of their paper and removing the connection to three-stage rotors. In this case, they might leave out Section 2 and focus solely on their design approach for single stage turbines. This would include changing the title of the paper and adjusting the content of the paper accordingly.

We greatly appreciate the honorable reviewer for constructive suggestions. Section 2 gives our contribution in the present work, the proposed design of floating Darrieus-type wind turbines with three-stage rotors solved the problem of starting large turbines, because in our design the rotor is subdivided into three stages with a different radius and height, which means a different AR. This configuration allows us to vary the output performance, also by using two-bladed design, we can integrate a mechanism that allows varying the rotor radius according to the wind velocity available.

4.2 Text - Description of design approach

In both cases, the description of the design approach, i.e. the discussion of Fig. 7, needs to be reworked. From Fig. 7 it is not clear, what the respective components of QBlade do. The tool XFoil is tyically used for 2D airfoil analysis. It is unclear, how the results of Xfoil are used for a subsequent blade design. The used software modules must be described in much more detail and connected to the analyses of Section 3 and 4.

Following this recommendation formulated by the reviewer, we have adjusted the descriptions of software modules inside QBlade as follows (lines 221-232):

Figure 7. Software modules inside QBlade.

QBlade is used as an open source framework for the design and aerodynamic simulations of the wind turbines. In a recent paper by Nachtane et al. [34] QBlade solver was used to predict the hydrodynamic performance of a new hydrofoil named NTSXX20. QBlade utilizes the Blade Element Momentum (BEM) method for the simulation of HAWTs, and a Double Multiple Stream tube (DMST) model for the VAWTs performance. The XFOIL code is integrated seamlessly into QBlade based on the viscous-inviscid coupled panel method to generate 2D airfoil coordinates for blade design, airfoil lift, drag coefficients and 360° polar extrapolation for turbine simulations [35]; Figure 7 shows the different modules and types of analysis (shown inside the dotted square) that can be performed by Qblade for aerodynamic simulation.

4.3 Text- Other

  • Provide a source for the 59.3% power of Betz' limit (p. 2)

Thank you for this suggestion. We have cited reference [7] in the text, and also in the reference section as follows (lines 48-404):

  1. Burton T, Sharpe D, Jenkins N, Bossanyi E. Wind energy handbook, Chichester: John Wiley and Sons Ltd; England .2001; pp. 45.

  • 6a): Why is this Figure shown? It is unclear which part of the text it refers to.

In the literature [25], many aerodynamic models for VAWT have been proposed, such as Double Multiple Stream Tube (DMST), Multiple Stream Tube (MST), Vortex, Cascade, Panel, etc. Each of them has its advantages and disadvantages, Table 1 gives a comparison of the aerodynamic models according to important criteria such as: complexity, computational time, etc. Therefore, it is necessary to find a compromise between the validity of the results, the problem complexity, and the computational effort, depending on the objectives fixed. The model chosen for this work is the DMST developed by Paraschivoiu in 1981 [32]. This model combined the MST model with the double actuator disk theory (see Figure 6a) to predict the performance of VAWT. The advantages of this model can be seen through the fact that it is relatively simple to implement and gives better correlation between calculated and experimental results [25]. However, the major drawback of this model is to give over prediction of power for a high solidity turbine and there appears to be a convergence problem for the same type of turbine, especially in the downstream side and at the higher tip speed ratio [29].

Figure 6. (a) Shematic of two-actuator disk in tandem (b) Diagrammatic of DMST model.

As shown in Figure 6b, the rotor's area is divided into a set of adjacent parallel stream tubes. The calculation is performed separately for the upwind and downwind half cycles, the blade element theories and the momentum conservation are used for calculating the aerodynamic forces acting on the blades, which then makes it possible to calculate the torque and the power generated.

  • "Advantages and disadvantages" of aerodynamic models on p. 7 must be stated more clearly

Thank you for this recommendation. We have added table 2 which summarizes the advantages and disadvantages of different aerodynamic modules (lines 194-199).

The advantages of this model can be seen through the fact that it is relatively simple to implement and gives better correlation between calculated and experimental results [25]. However, the major drawback of this model is to give over prediction of power for a high solidity turbine and there appears to be a convergence problem for the same type of turbine, especially in the downstream side and at the higher tip speed ratio [29].

  • 9: What are these parameters based on? What is the stage solidity? Which stage is analyzed in the following?

Thanks for this valuable suggestion by the reviewer. This comment is now considered in revised paper and we have added Table 3 and Table 4 to give more detail about input parameters for aerodynamic simulations. The descriptions are revised in the subsection Input parameters used for the aerodynamic simulation (lines 281-284) and copied below:

Table 3: Influential geometric parameters for aerodynamic simulations.

  • TSR greater than 3.5 are mentioned on p. 9 but not shown in Fig. 8.

The correction has been made. We have added more details about the Qblade software validation as follows (lines 233-260):

In order to show the validity of this software, the Qblade was compared with the experimental and computational results of Raciti Castelli et al. [36] with various TSRs. The comparison of the power coefficient versus TSR results from the QBlade with experimental and numerical results are shown in Figure 8. It can be observed that the general trend of the experimental plot was captured by the Qblade where the quantitative differences (presented in Table 2) are found to be the largest for the highest TSR equal to 2.21. The observed deviations may be explained by the uncertainties associated with the experimental data, e.g. boundary conditions, the blade surface roughness, the wake-blade interaction. Also, the effect of blade-spoke connection on the  of the turbine is known to be more significant in higher TSRs [37]. As well as Qblade overestimates the  at high TSRs due to the dynamic stall effects are not considered, which is difficult to predict because the angle of attack change rapidly and the turbulence generated. Another source of error is from the extrapolation of the initial data obtained from XFOIL as the airfoil characteristics are required for 360˚. Regarding the extrapolation, Montgomerie [38, 39] and Viterna [40] are the most widely used methods. The Montgomery method is founded on the assumption as a thin plate, whereas the Viterna method is formulated on the basis of the potential flow theory. Extrapolated values provide a good estimate if the initial limited data are obtained from experiments. However, in this the case the initial data is an over predicted by Xfoil which can further extend the high lift/drag ratio error for other angle of attack. Furthermore, the Montgomery and Viterna methods are intended for static airfoils, which did not consider the unsteady effects. The airfoil characteristics calculated from the Xfoil assume that the flow is laminar, but in fact, VAWTs airfoils face a turbulent and unstable incoming flow, and it becomes worse in the downstream side, where the blade operates entirely in wake for a high solidity turbine [41].

Figure 8. Comparison of power coefficient versus experiment and numerical results by Raciti Castelli et al. [36].

Table 2 presents percent differences between numerical and experimental results of the power coefficient. The relative deviation between the numerical and experimental results is given by the equation:

Table 2: Comparison of  from experiment, CFD, Qblade and their relative deviation

Thank you for this question, this difference is normal because we did not take the same boundary condition (airfoil, R, h, c, free stream velocity, Re).

4.3.1 Figures

One of the blades in Fig. 6 b) is illustrated upside-down

Thank you for this suggestion. We have improved the quality of Figure 6 and we eliminated one of the blades illustrated in upside-down (Figure 6(b)), as you can see below (lines 186-187):

Figure 6. (b) Diagrammatic of DMST model.

4.3.2 Equations

  • The density is missing in the definition of the Reynolds number (Eqn. 15)

Thank you for this suggestion. we checked equation 1 (lines 181-184):

Reynolds number is used to characterize the flow regime perceived by the blades. It represents the ratio between inertial forces and viscous forces:

  • In the definitions of forces acting on the blade (Eqns. 8, 9, 10), the velocity V must be V^2

Thanks for reviewers’ guidance. According to references [22,26,27], we checked equations (Eqns. 8, 9, 10) (lines 155-160):

The overall normal and tangential forces are given by:

4.4 Style

4.4.1 Figures

 The overall quality of the Figures is very low. Figures 1, 2, 4, 6 must be reworked because the text in the PDF is either not legible, too small or symbols are not printed or printed on top of each other in the PDF.

Thanks for your kind suggestion, we have reconstructed all the figures in Scalable Vector Graphics (SVG) format.

  • Equations 10, 17, 18 appear twice in the document.

Thank you for this suggestion. We eliminated the doubled equation in the revised version of the manuscript.

  • Equations 12, 13 are not legible in the document.

Thank you for this suggestion, we corrected equation 13, it can be calculated as suggested by Strickland [28] (lines 169-171):

  • TSR and a must be defined before Eqn. 4.

Thank you for this suggestion. We have made the modification you requested (Lines 140-141) as follows:

Tip speed ratio is defined as:

Where R and ω are the radius and the rotational speed of the rotor, respectively

  • Typesetting (indices and exponents) in the presentation of equations and units should be corrected

Thank you for this suggestion. We have made the modifications you requested in the revised manuscript.

4.4.3 Abbreviations

  • Some abbreviations are either not defined (e.g. FAWT) or defined twice (e.g. DMST).

Thank you for this suggestion. We have made the modifications you requested in the revised manuscript.

4.4.4 Text

  • At multiple occasions in the text, lists are concluded with "...". This should be avoided.

Thank you for this suggestion. We have made the modifications you requested in the revised manuscript.

  • The keyword "section" is not capitalized consistently.

Following this suggestion of the reviewer, we have corrected the capitalization error of the letter S (line 96).

Reviewer 3 Report

Title: Aerodynamic simulations for floating Darrieus-type wind turbines with three-stage rotors

Authors: Mohamed Amine Dabachi, Abdellatif Rahmouni, Eugen Rusu and Otmane Boukosour

The authors have analyzed 3 stage floating straight bladed VAWT using double multiple stream tube method (Qblade code). The manuscript cannot be accepted in the present form as some of technical have to clarified to see the methodology is acceptable or not.

  1. My main objection for the work is low solidity, 0.3. does DMST method work for low solidities. Based on my experience, for low solidity of 0.3 this method will not work
  2. The justification for validation is really bad. It is better that the authors look for experimental data whose solidity is close to what they would like to work on like 0.3 and use that data for validation.

Author Response

ANSWERS TO THE REVIEWERS' COMMENTS 

The specific corrections operated according to the suggestions of the reviewers are given next together with detailed explanations.

Respected Editor and Reviewers!

We greatly appreciate the valuable comments made by the reviewers. These comments are very valuable and helpful for revising and improving our paper, as well as for the guidance of our future research.

After carefully studying the comments, we have made corresponding changes, which are highlighted with ''track changes'' in the revised manuscript. Next are provided the answers and revisions we have made in responding to the reviewers’ comments on an item-by-item basis.

-----------------------------------------------------------------------------------------------------------------

Reviewer # 3:

Title: Aerodynamic simulations for floating Darrieus-type wind turbines with three-stage rotors

The authors have analyzed 3 stage floating straight bladed VAWT using double multiple stream tube method (Qblade code). The manuscript cannot be accepted in the present form as some of technical have to clarified to see the methodology is acceptable or not.

  1. My main objection for the work is low solidity, 0.3. does DMST method work for low solidities. Based on my experience, for low solidity of 0.3 this method will not work.

We gratefully acknowledge the reviewer for the acceptance to review our work, due to his experience in the field of VAWT. According to the literature review, we found that some research used DMST or Multiple StreamTube methods (MST) (the MST model is not able to differentiate between the up-wind and down-wind side of the turbine, however, the calculation for the DMST is done separately for the up-wind and down-wind side, and gives good results compared to MST).  

Please find bellow some references where the above approach have been used.

#Reference 1:

[1]  A. Meana-Fernández, J. M. F. Oro, K. M. A. Díaz, M. Galdo-Vega, and S. Velarde-Suárez, “Aerodynamic Design of a Small-Scale Model of a Vertical Axis Wind Turbine,” Proceedings, vol. 2, no. 23, p. 1465, Nov. 2018, doi: 10.3390/proceedings2231465.

Meana-Fernández et al. [1] applied the DMST model on the DU-06-W-200 profile found that the optimal solidity was ? = 0.333, the same as that which we found in our results.

#Reference 2:

Brusca et al. [2]: S. Brusca, R. Lanzafame, and M. Messina, “Design of a vertical-axis wind turbine: how the aspect ratio affects the turbine’s performance,” Int J Energy Environ Eng, vol. 5, no. 4, pp. 333–340, Dec. 2014, doi: 10.1007/s40095-014-0129-x.

‘’ From the graph provided in Fig. 1, the solidity which maximizes the power coefficient ? = 0.3 can be identified’’, the same as that which we found in our results.

In this work, the solidity varying between 0.1 and 0.75 as shown in Fig.1 below

Screenshot Figure 6 of the reference [2]

#Reference 3: Strickland, J.H., ‘’The Darrieus Turbine: A Performance Prediction Model Using Multiple Streamtubes’’, Sandia National Lanoratories, Report SAND75-0431, October 1975.

" Figures 6 and 7 depict the effect of solidity at blade Reynold’s numbers"

In this work, the solidity varying between 0.1 and 0.4 as shown in Figures below:

  1. The justification for validation is really bad. It is better that the authors look for experimental data whose solidity is close to what they would like to work on like 0.3 and use that data for validation.

Thank you very much for reviewer’s guidance, we greatly appreciate the honorable reviewer suggestion. In the literature, we only have this information. It is for this reason that the comparison between experimental and numerical data was made only for NACA0021. We validated the Qblade software with other experimental and numerical results to show the validity of the software. Once we showed this, a parametric study was performed by changing the solidity, number of blades, radius...).

Nevertheless, taking into account the above observation formulated by the reviewer, we have added a section related to the validation of Qblade software in the revised manuscript as follows (lines 221– 260):

QBlade is used as an open source framework for the design and aerodynamic simulations of the wind turbines. In a recent paper by Nachtane et al. [34] QBlade solver was used to predict the hydrodynamic performance of a new hydrofoil named NTSXX20. QBlade utilizes the Blade Element Momentum (BEM) method for the simulation of HAWTs, and a Double Multiple Stream tube (DMST) model for the VAWTs performance. The XFOIL code is integrated seamlessly into QBlade based on the viscous-inviscid coupled panel method to generate 2D airfoil coordinates for blade design, airfoil lift, drag coefficients and 360° polar extrapolation for turbine simulations [35]; Figure 7 shows the different modules and types of analysis (shown inside the dotted square) that can be performed by Qblade for aerodynamic simulation.

In order to show the validity of this software, the Qblade was compared with the experimental and computational results of Raciti Castelli et al. [36] with various TSRs. The comparison of the power coefficient versus TSR results from the QBlade with experimental and numerical results are shown in Figure 8. It can be observed that the general trend of the experimental plot was captured by the Qblade where the quantitative differences (presented in Table 2) are found to be the largest for the highest TSR equal to 2.21. The observed deviations may be explained by the uncertainties associated with the experimental data, e.g. boundary conditions, the blade surface roughness, the wake-blade interaction. Also, the effect of blade-spoke connection on the  of the turbine is known to be more significant in higher TSRs [37]. As well as Qblade overestimates the  at high TSRs due to the dynamic stall effects are not considered, which is difficult to predict because the angle of attack change rapidly and the turbulence generated. Another source of error is from the extrapolation of the initial data obtained from XFOIL as the airfoil characteristics are required for 360˚. Regarding the extrapolation, Montgomerie [38, 39] and Viterna [40] are the most widely used methods. The Montgomery method is founded on the assumption as a thin plate, whereas the Viterna method is formulated on the basis of the potential flow theory. Extrapolated values provide a good estimate if the initial limited data are obtained from experiments. However, in this the case the initial data is an over predicted by Xfoil which can further extend the high lift/drag ratio error for other angle of attack. Furthermore, the Montgomery and Viterna methods are intended for static airfoils, which did not consider the unsteady effects. The airfoil characteristics calculated from the Xfoil assume that the flow is laminar, but in fact, VAWTs airfoils face a turbulent and unstable incoming flow, and it becomes worse in the downstream side, where the blade operates entirely in wake for a high solidity turbine [41].

Figure 8. Comparison of the power coefficient versus experiment and numerical results by Raciti Castelli et al. [36].

Table 2 presents in percentage the differences between the numerical and experimental results of the power coefficient. The relative deviation between the numerical and experimental results is given by the equation:

Table 2: Comparison of  from experiment, CFD, Qblade and their relative deviation

Reviewer 4 Report

Figure 1 – very badly  formatted

Figure 2 – badly formatted, legend out of place

Eq. 1 – must be verified

Eq 3   ,  and not the way it is presented; the corrections in eq 1 must be considered

Eq. 5 – tanα bearing  figure  4,  is Eq. 5 correct?

Eq 13 – badly formatted

Line 184 – badly placed

Fig 6 – badly formatted

Eq. 17= Eq. 3

Eq7=Eq.20

Author Response

ANSWERS TO THE REVIEWERS' COMMENTS 

The specific corrections operated according to the suggestions of the reviewers are given next together with detailed explanations.

Respected Editor and Reviewers!

We greatly appreciate the valuable comments made by the reviewers. These comments are very valuable and helpful for revising and improving our paper, as well as for the guidance of our future research.

After carefully studying the comments, we have made corresponding changes, which are highlighted with ''track changes'' in the revised manuscript. Next are provided the answers and revisions we have made in responding to the reviewers’ comments on an item-by-item basis.

-----------------------------------------------------------------------------------------

Round 2

Reviewer 1 Report

The document has been improved and all my comments have been considered. The new section for the model validation has cleared my biggest concern. Maybe the language can be polished a bit more, especially in the new sections.

Reviewer 2 Report

I am still concerned about the mismatch between the conducted analysis and the title/overall goal of the paper. However, I do understand that in the given time frame the authors are unable to conduct a full 3D analysis of the problem. They have improved the quality of the presentation and added enough of the missing information to enable a publication of the manuscript.

Yet, there is one minor error that needs to be corrected before publication: In the definition of Reynolds number (Eqn. 17), the density has been added only in one of the terms and is still missing in the other terms. Also, the definition given here is valid only if TSR=1 (as c*V_inf/mu = TSR*V_inf*c/mu). The authors should check this derivation again.

Reviewer 3 Report

It can be seen that the authors has modified the manuscript according to the suggestions given by the reviewers. I think the manuscript can be accepted in the present form.

Reviewer 4 Report

No special recommandations